# Adaptive Sliding Mode Disturbance Observer and Deep Reinforcement Learning Based Motion Control for Micropositioners

**DOI:** 10.3390/mi13030458

**Published:** 2022-03-17

**Authors:** Shiyun Liang, Ruidong Xi, Xiao Xiao, Zhixin Yang

**Affiliations:** 1State Key Laboratory of Internet of Things for Smart City and Department of Electromechanical Engineering, University of Macau, Macau 999078, China; mb95407@um.edu.mo (S.L.); yb57466@umac.mo (R.X.); 2Department of Electronic and Electrical Engineering, Southern University of Science and Technology, Shenzhen 518055, China; xiaox@sustech.edu.cn

**Keywords:** micropositioners, reinforcement learning, disturbance observer, deep deterministic policy gradient

## Abstract

The motion control of high-precision electromechanitcal systems, such as micropositioners, is challenging in terms of the inherent high nonlinearity, the sensitivity to external interference, and the complexity of accurate identification of the model parameters. To cope with these problems, this work investigates a disturbance observer-based deep reinforcement learning control strategy to realize high robustness and precise tracking performance. Reinforcement learning has shown great potential as optimal control scheme, however, its application in micropositioning systems is still rare. Therefore, embedded with the integral differential compensator (ID), deep deterministic policy gradient (DDPG) is utilized in this work with the ability to not only decrease the state error but also improve the transient response speed. In addition, an adaptive sliding mode disturbance observer (ASMDO) is proposed to further eliminate the collective effect caused by the lumped disturbances. The micropositioner controlled by the proposed algorithm can track the target path precisely with less than 1 μm error in simulations and actual experiments, which shows the sterling performance and the accuracy improvement of the controller.

## 1. Introduction

Micropositioning technologies based on smart materials in precision industries have gained much attention for numerous potential applications in optical steering, micro-assembly, nano-inscribing, cell manipulation, etc. [1,2,3,4,5,6,7]. One of the greatest challenge in this research field is the uncertainties produced by various factors such as the dynamic model, environmental temperature, sensors performance, and the actuators’ nonlinear characteristics [8,9], which make the control of micropositioning system a demanding problem.

To address the uncertain problem, different kinds of control approach have been developed, such as the PID control method [10], sliding mode control [11,12], and adaptive control [13]. In addition, many researchers have integrated these control strategies to further improve the control performance. Victor et al. have proposed a scalable field-programmable gate array-based motion control system with a parabolic velocity profile [14]. A new seven-segment profile algorithm was developed by Jose et al. to improve the performance of the motion controller [15]. Combined with the backstepping strategy, Fei et al. proposed an adaptive fuzzy sliding mode controller in [16]. Based on the radial basis function neural network (RBFNN) and sliding mode control (SMC), Ruan et al. developed a RBFNN-SMC for nonlinear electromechanical actuator systems [17]. Gharib et al. designed a PID controller with a feedback linearization technique for path tracking control of a micropositioner [18]. Nevertheless, the performance and robustness of such model-based control strategies are still limited by the precision of the dynamics model. On the other hand, a sophisticated system model frequently leads to a complex control strategy. Although many researchers have considered the factors of uncertainties and disturbances, it is still difficult for the system to provide a precise and comprehensive process.

As the rapid development in artificial intelligence in recent years has roundly impacted the traditional control field, learning-based and data-driven approaches, especially reinforcement learning (RL) and neural networks, have become a promising research tropic. Different from traditional control strategies that need to make assumptions based on the dynamics model [19,20], reinforcement learning can directly learn the policy by interacting with the system. Back in 2005, Adda et al. presented a reinforcement learning algorithm for learning control of stochastic micromanipulation systems [21]. Li et al. designed a state–action–reward–state–action (SARSA) method using linear function approximation to generate an optimal path by controlling the choice of the micropositioner [22]; however, the reinforcement learning algorithms such as Q-learning [23] and SARSA [24] utilized in the aforementioned works are unable to deal with complex dynamics problems, especially the continuous state action space problem. With the spectacular improvement enjoyed by deep reinforcement learning (DRL), primarily driven by deep neural networks (DNN) [25], the DRL algorithms, such as the deep Q network (DQN) [26], policy gradient (PG) [27], deterministic policy gradient (DPG) [28], and deep deterministic policy gradient (DDPG) [29] with the ability to approximate the value function, have played an important role in continuous control tasks.

Latifi et al. introduced a model-free neural fitted Q iteration control method for micromanipulation devices; in this work, the DNN is adopted to represent Q-value function [30]. Leinen introduced the concept of experience playback in DQN and the approximate value function of the neural network into the SARSA algorithm for the control of a scanning probe microscope [31]. Both simulation and real experimental results have shown that their proposed RL algorithm based on the neural network could achieve better performance compared to traditional control methods to some extent; however, due to the collective effects of disturbances generated from nonlinear systems and deviations in value functions [29,32,33], the RL control method could induce significant inaccuracies in the tracking control tasks [34]. To improve the anti-disturbance capability and control accuracy, disturbance rejection control [35], time-delay estimation based control [36], disturbance observer-based controllers [37,38] have been proposed successively. To deal with this issue, a deep reinforcement learning controller integrated with an adaptive sliding mode disturbance observer (ASMDO) is developed in this work. Previous research on trajectory tracking control of DRL has shown that apparent state errors have always existed [39,40,41,42]. One of the main reasons is the inaccurate estimation of the action value function in DRL structure. As indicated in [43], even in elementary control tasks, accurate action values cannot be attained from the same action value function; therefore, in this work, the DDPG algorithm is developed with an integral differential compensator (DDPG-ID) added to cope with this situation. In addition, the comparison of the reinforcement learning control method with various common state-of-the-art control methods are listed in Table 1, which shows the pros and cons of these different methods.

In this study, deep reinforcement learning is leveraged into a novel optimal control scheme for complex systems. An anti-disturbance, stable, and precise control strategy is proposed for the trajectory tracking task of the micropositioner system. The contribution of this works are presented as follows:(1)A DDPG-ID algorithm based on deep reinforcement learning is introduced as a basic micropositioner system motion controller, which avoided the limitation of traditional control strategies to the accuracy and comprehensiveness of the dynamic model;(2)To eliminate the collective effect caused by the lumped disturbances from the micropositioner system and inaccurate estimation of the value function in deep reinforcement learning, an adaptive sliding mode disturbance observer (ASMDO) is proposed;(3)An integral differential compensator is introduced in DDPG-ID to compensate for the feedback state of the system, which improves the accuracy and response time of the controller, and further improves the robustness of the controller subject to external disturbances.

The manuscript is structured as follows. Section 2 presents the system description of the micropositioner. In Section 3, we develop a deep reinforcement learning control method combined with ASMDO and compensator, and parameters of the DNNs are illustrated. Then, simulation parameters and tracking results are given in Section 4. To further evaluate the performance of the proposed control strategy in the micropositioner, tracking experiments are presented in Section 4. Lastly, conclusions are given in Section 5.

## 2. System Description

The basic structure of micropositioner is shown in Figure 1, which consists of a base, a platform, and a kinematic device. The kinematic device is composed with an armature, an electromagnetic actuator, and a chain mechanism driven by electromagnetic actuator. As shown in Figure 1, there are mutual-perpendicular compliant chains actuated by the electron-magnetic actuator (EMA) in the structure. The movement of the chain mechanism is in accordance with the working air gap *y*. The EMA generates the magnetic force $T_{m}$, which can be approximated as:(1)$$T_{m}=k{\left(\frac{I_{c}}{y+p}\right)}^{2}$$
where *k* and *p* are constant parameters related to the electronmagnetic actuator, $I_{c}$ is the excitation current, and *y* is the working air gap between the armature and the EMA. Then, the electrical model of the system can be given as:(2)$$V_{i}=RI_{c}+\frac{d}{dt}\left(HI_{c}\right)$$
where $V_{i}$ is the input voltage from the EMA, *R* is the resistance of the coil and *H* denotes the coil inductance, which can be given as:(3)$$H=H_{1}+\frac{pH_{0}}{y+p}$$
where $H_{1}$ is the coil inductance while the air gap is infinite, and $H_{0}$ is the incremental inductance when the gap is zero. The motion equation for the micropositioner can be expressed as:(4)$$m\frac{d^{2}y}{dt^{2}}=\iota \left(\alpha_{0}-y\right)-T_{m}$$
where ι is the stiffness along the motion direction in the system, and $\alpha_{0}$ is the initial air gap.

According to Equations (1)–(4), they define $x_{1}=y$, $x_{2}=\dot{y}$, $x_{3}=I_{c}$ as the state variables and the control input $u=V_{i}$. Then, the dynamics model of the electromagnetic actuator can be written as:(5)$$\left\{\begin{array}{c} {\dot{x}}_{1}=x_{2}\\ {\dot{x}}_{2}=\frac{\iota }{m}\left(\alpha_{0}-x_{1}\right)-\frac{k}{m}{\left(\frac{x_{3}}{x_{1}+p}\right)}^{2}\\ {\dot{x}}_{3}=\frac{1}{H}\left(-Rx_{3}+\frac{H_{0}px_{2}x_{3}}{{\left(x_{1}+p\right)}^{2}}+u\right) \end{array}\right.$$

Define the variables $z_{1}=x_{1}$, $z_{2}=x_{2}$, $z_{3}=\frac{\iota }{m}\left(\alpha_{0}-x_{1}\right)-\frac{k}{m}{\left(\frac{x_{3}}{x_{1}+p}\right)}^{2}$, then we have
(6)$$\left\{\begin{array}{c} {\dot{z}}_{1}=z_{2}\\ {\dot{z}}_{2}=z_{3}\\ {\dot{z}}_{3}=f\left(x\right)+g\left(x\right)u \end{array}\right.$$
where $f\left(x\right)=-\frac{\iota x_{2}}{m}+\frac{2kx_{3}^{2}}{m{\left(x_{1}+p\right)}^{2}}\left(\frac{H\left(x_{1}+p\right)-pH_{0}}{H{\left(x_{1}+p\right)}^{2}}x_{2}+\frac{R}{H}\right)$, $g\left(x\right)=-\frac{2kx_{3}}{Hm{\left(x_{1}+p\right)}^{2}}$, and $z_{1}$ is the system output.

In realistic engineering application, there always exist some uncertainties of the system, then system Equation (6) can be rewritten as:(7)$$\left\{\begin{array}{c} {\dot{z}}_{i}=z_{i+1},i=1,2\\ {\dot{z}}_{3}=f_{0}\left(x\right)+g_{0}\left(x\right)u+\left(\Delta f\left(x\right)+\Delta g\left(x\right)u\right)+d \end{array}\right.$$
where $f_{0}\left(x\right)$ and $g_{0}\left(x\right)$ denote the nominal part of the micropositioner system and Δf(x), Δg(x) denote the uncertainties of the modeling system; *d* denotes the external disturbances. Then, defining D=(Δf(x)+Δg(x)u)+d, we have
(8)$$\left\{\begin{array}{c} {\dot{z}}_{i}=z_{i+1},i=1,2\\ {\dot{z}}_{3}=f_{0}\left(x\right)+g_{0}\left(x\right)u+D \end{array}\right.$$
where *D* is the lumped system disturbances. The following assumption is exploited [44]:

**Assumption** **1.**
*The lumped interference D is bounded and its upper bound is less than a fixed parameter $\beta_{1}$ and the derivative of D is unknown but bounded.*


**Remark** **1.**
*Assumption 1 is reasonable since all micropositioner platforms are accurately designed and parameter identified, and all disturbances are remained in a controllable domain.*


## 3. Design of ASMDO and DDPG-ID Algorithm

In this section, the adaptive sliding mode disturbance observer (ASMDO) is introduced based on the dynamics of the micropositioner. Then, the DDPG-ID control method and pseudocode are given.

### 3.1. Design of Adaptive Sliding Mode Disturbance Observer

To develop the ASMDO, a virtual dynamic is firstly designed as
(9)$$\left\{\begin{array}{c} {\dot{\eta }}_{i}=\eta_{i+1},i=1,2\\ {\dot{\eta }}_{3}=f\left(z\right)+g\left(z\right)u+\hat{D}+\rho \end{array}\right.$$
where $\eta_{i},i=1,2,3$ are auxiliary variables, $\hat{D}$ is the estimation of lumped disturbances, ρ denotes the sliding mode term, which is introduced afterwards.

Define a sliding variable $S=\sigma_{3}+k_{2}\sigma_{2}+k_{1}\sigma_{1}$, where $\sigma_{i}=x_{i}-\eta_{i},i=1,2,3$, $k_{1}$ and $k_{2}$ are positive design parameters. Then the sliding mode term ρ is designed as
(10)$$\rho =\lambda_{1}S+k_{2}\sigma_{3}+k_{1}\sigma_{2}+\lambda_{2}\mathrm{sgn}\left(S\right)$$
where $\lambda_{1}$, $\lambda_{2}$ are positive design parameters with $\lambda_{2}\geq \beta_{1}$.

Choosing an unknown constant $\beta_{2}$ to present the upper bound of $\dot{D}$, the ASMDO is proposed as:(11)$$\dot{\hat{D}}=k\left({\dot{x}}_{3}-f_{0}\left(z\right)-g_{0}\left(z\right)u-\hat{D}\right)+\left({\hat{\beta }}_{2}+\lambda_{3}\right)\mathrm{sgn}\left(\rho \right)$$
where *k* and $\lambda_{3}$ are positive design parameters and ${\hat{\beta }}_{2}$ is defined as the estimation of $\beta_{2}$ given by ${\dot{\hat{\beta }}}_{2}=-\delta_{0}{\hat{\beta }}_{2}+∥\rho ∥$, with $\delta_{0}$ is a small positive number.

Then, the output $\hat{D}$ of the ASMDO is used as a compensation of the control input to eliminate the uncertainties generated by the system and external disturbances.

**Remark** **2.**
*Choosing $V_{1}=\frac{1}{2}S^{2}$ and $V_{2}=\frac{1}{2}\left({\tilde{D}}^{2}+{\tilde{\beta }}_{2}^{2}\right)$, where $\tilde{D}=D-\hat{D}$, ${\tilde{\beta }}_{2}=\beta_{2}-{\hat{\beta }}_{2}$ as two Lyapunov function, derivative $V_{1}$ and $V_{2}$ with respect to time, it is easy to prove that both S and $\tilde{D}$ will exponentially converge to the equilibrium point, so the proof process is not repeated.*


### 3.2. Design of DDPG-ID Algorithm for Micropositioner

The goal of reinforcement learning is to obtain a policy for the agent that could maximizes the cumulative reward through interactions with the environment. The environment is usually formalized as a Makov decision process (MDP) described by a four-tuple (S,A,P,R), where *S*, *A*, *P*, and *R* represent the state space of environment, set of actions, state transition probability function, and reward function separately. At each time step *t*, the agent in current state $s_{t}\in S$ takes action $a_{t}\in A$ from policy $\pi (a_{t}|s_{t})$, then the agent acquires a reward $r_{t}\leftarrow R(s_{t},a_{t})$ and enters the next state $s_{t+1}$ according to the state transition probability function $P(s_{t+1}|s_{t},a_{t})$. Based on the Markov property, the Bellman equation of action–value function $Q_{\pi }\left(s_{t},a_{t}\right)$, which is used for calculating the future expected reward, can be given as:(12)$$Q_{\pi }\left(s_{t},a_{t}\right)=E_{\pi }\left(r_{t}+\gamma Q_{\pi }\left(s_{t+1},a_{t+1}\right)\right)$$
where γ∈[0,1] denotes the discount factor.

In trajectory tracking control task of micropositioner, state $s_{t}$ is state array about the air gap *y* of micropositioner at time *t*. Action $a_{t}$ is the voltage *u* applied by the controller to micropositioner. As shown in Figure 2, DDPG is one of actor–critic algorithms, which has an actor and a critic. The actor is responsible for generating actions and interacting with the environment, and the critic evaluates the performance of the actor and guides the action in the next state.

The action–value function and policy approximation are parameterized by DNN to solve the continuous states and actions problem in micropositioner with $Q\left(s_{t},a_{t},w^{Q}\right)≐Q_{\pi }\left(s_{t},a_{t}\right)$, $\pi_{w^{\mu }}\left(a_{t}|s_{t}\right)≐\pi \left(a_{t}|s_{t}\right)$, where $w^{Q}$ and $w^{\mu }$ are the parameters of neural networks in action–value function and policy function. Under the prerequisite of using the neural network approximation representation policy function, the neural network gradient update method is used to seek the optimal policy π.

DDPG-ID uses deterministic policy $\pi (s_{t},w^{\mu })$ rather than traditional stochastic policy $\pi_{w^{\mu }}\left(a_{t}|s_{t}\right)$, where the output of policy is the action $a_{t}$ with highest probability to current state $s_{t}$, $\pi \left(s_{t},w^{\mu }\right)=a_{t}$. The policy gradient is given as
(13)$$\nabla_{w^{\mu }}J\left(\pi \right)=\;E_{s∼\rho^{\pi \;}}[\left.\;\nabla {\;}_{w^{\mu }}\pi \left(s,w^{\mu }\right)\nabla {\;}_{a}Q\left(s,a,w^{Q}\right)\right]$$
where $J\left(\pi \right)=E_{\pi }\left[\sum_{t=1}^{T}\gamma^{\left(t-1\right)}r_{t}\right]$ is the expectation of discount accumulative rewards, *T* denotes the final time of a whole process, $\rho^{\pi }$ is the distribution of state following the deterministic policy. Value function $Q(s_{t},a_{t},w^{Q})$ is updated by calculating time temporal-difference error (TD-error), which can be defined as
(14)$$e_{TD}=r_{t}+\gamma Q\left(s_{t+1},\pi \left(s_{t+1}\right)\right)-Q\left(s_{t},a_{t}\right)$$
where $e_{TD}$ is the TD-error, $r_{t}+\gamma Q\left(s_{t+1},\pi \left(s_{t+1}\right)\right)$ represents the TD target value. By minimizing the TD-error, the parameters are updated backwards through the neural network gradient.

To avoid the convergence problem of single network caused by correlation between TD target value and current value [45,46], A target Q network $Q_{T}^{\prime }\left(s_{t+1},a_{t+1}^{\prime },w^{Q^{\prime }}\right)$ is introduced to calculate network portion of TD target value and an online Q network $Q_{O}\left(s_{t},a_{t},w^{Q}\right)$ is used to calculate current value in critic. Both these two DNN have the same structure. The actor also has an online policy network $\pi_{O}\left(s_{t},w^{\mu }\right)$ to generate current action and a target policy network $\pi_{T}\left(s_{t},w^{\mu^{\prime }}\right)$ to provide the target action $a_{t+1}^{\prime }$. $w^{\mu^{\prime }}$ and $w^{Q^{\prime }}$ separately represent the parameters of target policy and target Q networks.

In order to improve the stability and efficiency during RL training, experience replay technology is utilized in this work, which saves transition experience $\left(s_{t},a_{t},r_{t},s_{t+1}\right)$ into the experience replay buffer Ψ at each interaction with the environment for subsequent updates. In each training time *t*, a minibatch of *M* transitions $\left(s_{j},a_{j},r_{j},s_{j+1}\right)$ from the experience replay buffer are extracted to calculate the gradients and update neural networks.

An integral differential compensator is developed in deep reinforcement learning structure to improve the accuracy and responsiveness of tracking tasks in this work, which is shown in Figure 2. The integral portion of the state is utilized to increase the control input continuously, which would eventually reduce tracking error. The differential part is integrated to reduce the system oscillation and accelerates stability. The proposed compensator is designed as follows:(15)$$s_{ID}^{t}=y_{e}^{t}+\alpha \sum_{n=1}^{t}y_{e}^{t}+\beta \left(y_{e}^{t}-y_{e}^{t-1}\right)$$
where $s_{ID}^{t}$ represents the compensator error at time *t*, $y_{e}^{t}=\sqrt{{\left(y_{d}^{t}-{\hat{y}}^{t}\right)}^{2}}$, $y_{d}^{t}$ represents the desired trajectory at time *t*, ${\hat{y}}^{t}$ is the measured air gap at time *t* and $y_{e}^{t}$ is the error between them. α is the integral gain and β is the differential gain.

Then the state $s_{t}$ at time *t* can be described as:(16)$$s_{t}={\left[\begin{array}{ccccc} s_{ID}^{t} & {\hat{y}}^{t} & {\dot{\hat{y}}}^{t} & y_{d}^{t} & {\dot{y}}_{d}^{t} \end{array}\right]}^{T}$$
where ${\dot{\hat{y}}}^{t}$ and ${\dot{y}}_{d}^{t}$ represent the derivatives of ${\hat{y}}^{t}$ and $y_{d}^{t}$.

The reward $r_{t}$ function designed is to measure the tracking error:(17)$$\begin{array}{c} r_{t}=\left\{\begin{array}{c} -4,y_{e}^{t}>0.005\\ +5,0.003<y_{e}^{t}⩽0.005\\ +10,0.001<y_{e}^{t}⩽0.003\\ +18,y_{e}^{t}⩽0.001 \end{array}\right. \end{array}$$

As shown in Figure 3, the adaptive sliding mode disturbance observer (ASMDO) is embedded into the DDPG-ID between the actor and micropositioner system environment. Action $a_{t}$ with the environment is expressed as
(18)$$a_{t}=\pi_{O}\left(s_{t},w^{\mu }\right)+{\hat{D}}_{t}+N_{t}$$
where $w^{\mu }$ is the parameters of online policy network $\pi_{O}$, ${\hat{D}}_{t}$ is the estimation of the micropositioner system at time *t*, and $N_{t}$ is Gaussian noise for action exploration.

#### 3.2.1. Critic Update

After selecting *M* transitions $\left(s_{j},a_{j},r_{j},s_{j+1}\right)$ samples from experience replay buffer Ψ, the Q value is calculated. The online Q network is responsible for calculating the current Q value, which is as follows:(19)$$Q_{O}\left(s_{j},a_{j},w^{Q}\right)=w^{Q}\phi \left(s_{j},a_{j}\right)$$
where $\phi (s_{j},a_{j})$ represents the input of online Q network, which is an eigenvector consisting of state $s_{j}$ and action $a_{j}$.

The target Q network $Q_{T}^{\prime }$ is defined as:(20)$$Q_{T}^{\prime }\left(s_{j+1},\pi_{T}\left(s_{j+1},w^{\mu^{\prime }}\right),w^{Q^{\prime }}\right)=w^{Q^{\prime }}\phi \left(s_{j+1},\pi_{T}\left(s_{j+1},w^{\mu^{\prime }}\right)\right)$$
where $\phi (s_{j+1},\pi_{T}\left(s_{j+1},w^{\mu^{\prime }}\right))$ is the input of the target Q network, which is a eigenvector consisting state $s_{j+1}$ and target policy network output $\pi_{T}\left(s_{j+1},w^{\mu^{\prime }}\right)$.

For target policy network $\pi_{T}$, the equation is:(21)$$\pi_{T}\left(s_{j+1},w^{\mu^{\prime }}\right)=w^{\mu^{\prime }}s_{j+1}$$
Then, we rewrite the target Q value $Q_{T}$ as:(22)$$Q_{T}=r_{j}+\gamma Q_{T}^{\prime }\left(s_{j+1},\pi_{T}\left(s_{j+1},w^{\mu^{\prime }}\right),w^{Q^{\prime }}\right)$$
where $r_{j}$ is the reward from the selected samples.

Since *M* transitions $\left(s_{j},a_{j},r_{j},s_{j+1}\right)$ are sampled from experience buffer Ψ, the loss function of the update critic is shown in Equation (Equation 23).
(23)$$L\left(w^{Q}\right)=\frac{1}{M}\sum_{j=1}^{M}{\left(Q_{T}-Q_{O}\left(s_{j},a_{j},w^{Q}\right)\right)}^{2}$$
where $L\left(w^{Q}\right)$ is the loss value of critic.

In order to smooth the target network update process, the soft update is applied without copying parameters periodically as:(24)$$w^{Q^{\prime }}\leftarrow \tau w^{Q}+\left(1-\tau \right)w^{Q^{\prime }}$$
where τ is the update factor, usually a small constant.

The diagram of Q network is shown in Figure 4, which is a parallel neural network. The Q network includes both state and action portions, and the output value of Q network is based on state and action. The state portion of the neural network consists of a state input layer, three full connection layers, and two ReLU layers clamped between the three full connection layers. The neural network of the action portion contains an action input layer and a full connection layer. The output layers of the above two portions are combined entering the neural network of the common part, which contains a ReLU layer and one output layer.

The parameters of each layer in the Q network are shown in Table 2.

#### 3.2.2. Actor Update

The output of online policy network is
(25)$$\pi_{O}=w^{\mu }s_{j}$$

On account of using deterministic policy, the calculation of the policy gradient has no integrals of action *a*, but instead has the derivatives of the value function $Q_{O}$ with respect to action *a* in comparison with stochastic policy. The gradient formula can be rewritten as follows:(26)$$\nabla_{w^{\mu }}J\approx \frac{1}{M}\sum_{j}^{M}\left(\nabla_{a_{j}}Q_{O}\left(s_{j},aj,w^{Q}\right)\nabla_{w^{\mu }}\pi_{O}\left(s_{j},w^{\mu }\right)\right)$$
where the weights $w^{\mu }$ are updated with the gradient back-propagation method. The target policy network is also updated with soft update pattern as follows:(27)$$w^{\mu^{\prime }}\leftarrow \tau w^{\mu }+\left(1-\tau \right)w^{\mu^{\prime }}$$
where τ is the update factor, usually a small constant.

Figure 5 shows the diagram of the policy network in this paper, which contains a state input layer, a full connection layer, a tanh layer, and an output layer. The parameters of each layer in the policy network are shown in Table 3.

The Algorithm 1 pseudocode can be shown as:
**Algorithm 1** DDPG-ID Algorithm.  1:Randomly initialize online Q network with weights $w^{Q}$  2:Randomly initialize online policy network with weights $w^{\mu }$  3:Initialize the target Q network by $w^{Q^{\prime }}\leftarrow w^{Q}$  4:Initialize the target policy network by $w^{\mu^{\prime }}\leftarrow w^{\mu }$  5:Initialize the experience replay buffer Ψ  6:Load the simplified micropositioner dynamic model  7:**for** episode = 1, MaxEpisode **do**  8:    Initialize a noise process N for exploration  9:    Initialize ASMDO and ID compensator10:    Randomly initialize micropositioner states11:    Receive initial observation state $s_{1}$12:    **for** step = 1, *T* **do**13:       Select action $a_{t}=\pi_{O}\left(s_{t}\right)+{\hat{D}}_{t}+N_{t}$14:       Use $a_{t}$ to run micropositioner system model15:       Process errors with integral differential compensator16:       Receive reward $r_{t}$ and new state $s_{t+1}$17:       Store transition $\left(s_{t},a_{t},r_{t},s_{t+1}\right)$ in replay buffer Ψ18:       Randomly sample a minibatch of *M* transitions $\left(s_{j},a_{j},r_{j},s_{j+1}\right)$ from Ψ19:       Set $Q_{T}=r_{j}+\gamma Q_{T}^{\prime }\left(s_{j+1},\pi_{T}\left(s_{j+1},w^{\mu^{\prime }}\right),w^{Q^{\prime }}\right)$20:       Minimize loss: $L\left(\;w^{Q}\;\right)\;\;=\;\;\frac{1}{M}\;\;\sum_{j=1}^{M}\;{\left(\;Q_{T}-Q_{O}\;\left(\;s_{j},\;a_{j},\;w^{Q}\;\right)\;\right)}^{2}$ to update online Q network21:       Use the sampled policy gradient to update online policy network:       $\nabla_{w^{\mu }}J\;=\;\frac{1}{M}\;\sum_{j}^{M}\;\left(\nabla_{a_{j}}\;Q_{O}\left(s_{j},a_{j},w^{Q}\right)\nabla_{w^{\mu }}\pi_{O}\;\left(s_{j},w^{\mu }\right)\;\right)$22:       Update the target networks: $w^{Q^{\prime }}\leftarrow \tau w^{Q}+\left(1-\tau \right)w^{Q^{\prime }},w^{\mu^{\prime }}\leftarrow \tau w^{\mu }+\left(1-\tau \right)w^{\mu^{\prime }}$23:  **end for**24:**end for**

## 4. Simulation and Experimental Results

In this section, two kinds of periodic external disturbances were added to verify the practicability of the proposed ASMDO and three distinct desired trajectories were utilized to evaluate the performance of proposed deep reinforcement learning control strategy. An traditional DDPG algorithm and a well-tuned PID strategy were adopted for comparison. To further verify the spatial performances of the proposed algorithm, two kinds of different trajectories were introduced in the experiments.

### 4.1. Simulation Results

The parametric equations of two kinds of periodic external disturbances are defined as $d_{1}=0.1\sin\left(2\pi t\right)+0.1\sin\left(0.5\pi t+\frac{\pi }{3}\right)$, and $d_{2}=0.1+0.1\sin\left(0.5\pi t+\frac{\pi }{3}\right)$. Based on the micropositoner model proposed in [44], the effectiveness of the observer is presented in Figure 6 and Figure 7.

The disturbance estimation results from the proposed ASMDO are presented in Figure 6a and Figure 7a, it is can be seen that the observer could track the given disturbance rapidly. The estimation errors are less than 0.01 mm in Figure 6b and Figure 7b, which shows the effectiveness of the ASMDO as interference compensation.

The dynamics model of micropositioner is given in Section 2, and its basic system model parameters are from our previous research [44,47], which is shown in Table 4. The DDPG algorithm is defined in same neural network structure and training parameters as DDPG-ID in this paper. The training parameters of the DDPG-ID and DDPG are shown in Table 5.

The first desired trajectory designed for tracking control simulation is a waved signal. According to the initial conditions, the parametric equation of the waved trajectory is defined as:(28)$$y_{d}\left(t\right)=0.985-0.015\sin\left(\frac{\pi t}{4}-\frac{\pi }{2}\right)$$

The training process of both DDPG-ID and DDPG are run on the same model with stochastic initialized micropositioner states. During the training evaluation, a larger episode reward indicates a more accurate and lower error control policy. It is shown in Figure 8 that DDPG-ID reaches the maximum reward score with fewer episodes compared to DDPG, which reveals that DDPG-ID algorithm converge faster than DDPG algorithm. Comparing Figure 8a with Figure 8b, the average reward of DDPG-ID training process is larger than DDPG’s average reward in stable state, which further indicates that policy learned by DDPG-ID algorithm has better performance. The trained algorithms are employed for tracking control of micropositioner system simulation experiments.

The tracking results of the waved trajectory is shown in Figure 9. The RMSE value, MAX value, and mean value of the tracking errors for these three control methods are provided in Table 6. In terms of tracking accuracy, the trained DDPG-ID controller has a better performance compared to DDPG and PID, which has smaller state error and smoother tracking trajectory. The tracking error of the DDPG-ID algorithm ranges from $-8\times 10^{-4}$ to $9\times 10^{-4}$ mm, which is almost about a half of the DDPG policy. In the interim, the DDPG controller has a lesser tracking error than PID. A huge oscillation has been induced by the PID controller, which will affect the hardware to a certain extent in the actual operation process. This huge oscillation input signal is much larger than a normal control input signal, which typically ranges from 0 to 11 V. Based on the characteristics of reinforcement learning, it is hard for a well-trained policy to generate such a shock signal.

As can be seen in these figures, the tracking error of DDPG-ID in periodic trajectory is still less than the others, which ranges from $-1.6\times 10^{-4}$ to $9\times 10^{-4}$ mm. Similar to the previous waved trajectory, the control input based on DDPG has shown better performance in terms of oscillations.

Another tracking results of a periodic trajectory is illustrated in Figure 10, and the tracking errors comparison of these three control methods are given in Table 7. The parametric equation of the periodic trajectory is defined as
(29)$$y_{d}\left(t\right)\;=\;0.981-0.015\sin\left(\;\frac{\pi t}{4}-\frac{\pi }{2}\;\right)+0.008\sin\left(\;\frac{\pi t}{2}-\frac{\pi }{16}\;\right).$$

To further demonstrate the universality of the DDPG-ID policy, a periodic step trajectory is also utilized for comparison. The step signal with a period of 8 s is designed as the desired trajectory, which is shown in Figure 11a. The well-tuned PID controller is also tested in this step trajectory simulation. Since intense oscillations emerge, the results of PID show extremely worse performance are not shown in this paper.

According to Figure 11, the tracking result of DDPG-ID algorithm remains stable with the tracking error bounded in $-2\times 10^{-4}$ to $9\times 10^{-4}$ mm, which is still as a half of DDPG’s performance. Due to the characteristic of the step signal, the state error will become tremendous during the step transition. Errors of DDPG-ID and DDPG are observed dropping quickly after step transition. It can be seen from Table 8 that the errors of DDPG-ID algorithm are substantially less than that of DDPG algorithm. As to the control inputs, the value of DDPG still fluctuates considerably when the state converges stable.

According to above simulation results, it can be concluded that the control policy of DDPG-ID has triumphantly dealt with collective effect caused by disturbance and inaccurate estimation of deep reinforcement learning comparing to DDPG. The comparison results also have demonstrated the excellent control performance of the policy learned by DDPG-ID algorithm.

### 4.2. Experimental Results

The speed, acceleration, and direction of these designed trajectories vary with time, which makes the experiments results more trustworthy. In each test, the EMA in micropositioner is regulated for tracking the desired path of working air gap.

As shown in Figure 12, a laser displacement sensor is utilized to detect the motion states. Then DDPG-ID algorithm was administered through a SimLab board transplanted with Matlab-Simulink. The EMA controls the movement of the chain mechanism by executing the control signal, which is from the analog output port of SimLab board. The analog input port of SIMLAB board is connected with the signal output from the laser displacement sensor.

Figure 13 shows the tracking experiment results of the waved trajectory. It reaches the starting point on a straight track with a speed of 5.6 μm/s. At time 5 s, it begins to track the desired waved trajectory in three periods, and the waved trajectory can be described as $y_{d}\left(t\right)=28+25\sin\left(\frac{\pi t}{10}+\frac{\pi }{2}\right)$. The tracking error fluctuates within ±1.5μm, which is demonstrated in Figure 13b. Except for several particular points of time, the tracking errors could range from ±1 μm.

Another periodic trajectory tracking experiment was also executed. As shown in Figure 14, the desired periodic trajectory starts at time 5 s, and it is defined as $y_{d}\left(t\right)=35-25\sin\left(\frac{\pi t}{7.5}-\frac{2\pi }{3}\right)-5\sin\left(\frac{\pi t}{15}+\frac{\pi }{6}\right)$. The tracking error of the periodic trajectory still range from ±1.5μm.

The experimental results show that the proposed DDPG-ID algorithm is able to closely track above two trajectories. Compared with the simulation results, the tracking error does not increase significantly, and it can be maintained between −1 μm and +1 μm.

## 5. Conclusions and Future Works

In this paper, a composite controller is developed based on an adaptive sliding mode disturbance observer and a deep reinforcement learning control scheme. A deep deterministic policy gradient is utilized to obtain the optimal control performance. To improve the tracking accuracy and transient response time, an integral differential compensator is applied during the learning process in the actor–critic framework. An adaptive sliding mode disturbance observer is developed to further retrench the influence of modeling uncertainty, external disturbances, and the effect of inaccurate value function. In comparison with the existing DDPG and the most commonly used PID controller, the trajectory tracking results has successfully indicated the satisfactory performances and the precision of the control policy based on the DDPG-ID algorithm in the simulation. The tracking errors are less than 1 μm, which shows the significant tracking efficiency of the proposed methods. The experimental results also indicate the high accuracy and strong anti-interference capability of the proposed deep reinforcement learning control scheme. To further improve the tracking effect and realize micro-manipulation tasks in the future work, specific operation experiments will be performed such as cell manipulation, micro-assembly, etc.

## Figures and Tables

**Figure 1 micromachines-13-00458-f001:**
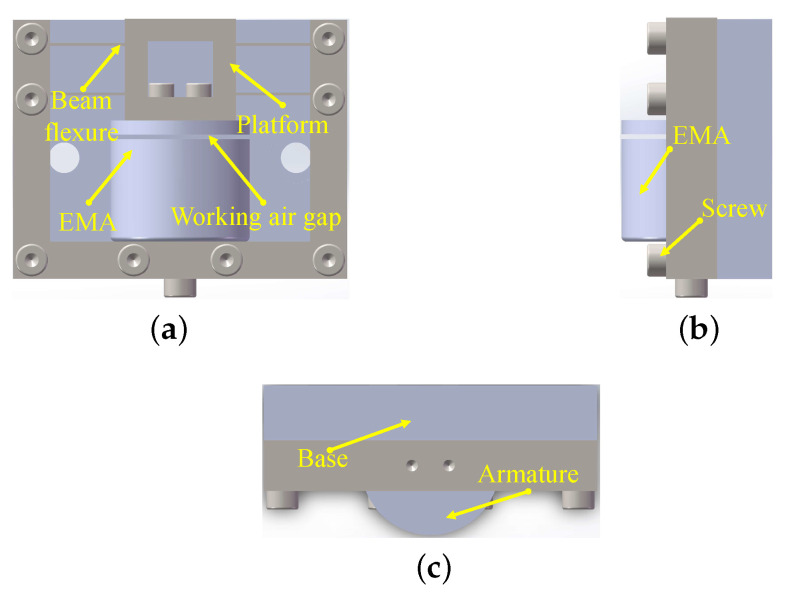
The diagrammatic model of EMA actuated micropositioner. (**a**) The front view of micropositioner. (**b**) The end view of micropositioner. (**c**) The vertical view of micropositioner.

**Figure 2 micromachines-13-00458-f002:**
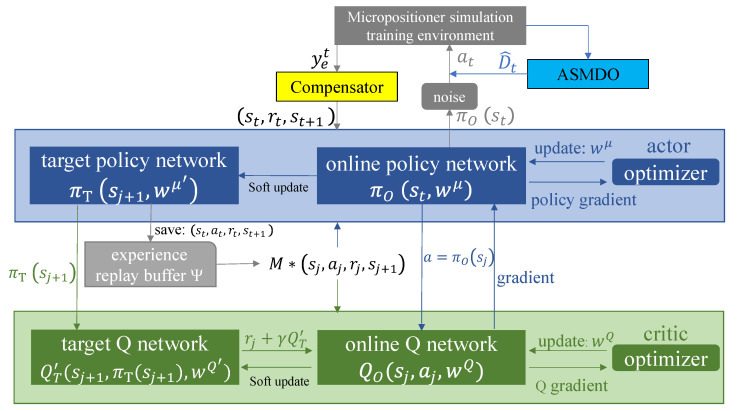
The structure diagram of DDPG-ID algorithm.

**Figure 3 micromachines-13-00458-f003:**
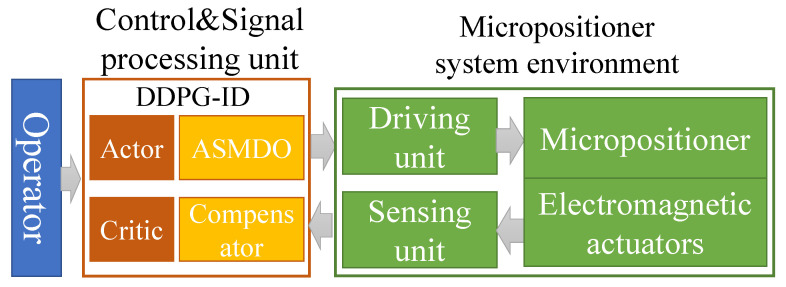
System signal flow chart.

**Figure 4 micromachines-13-00458-f004:**
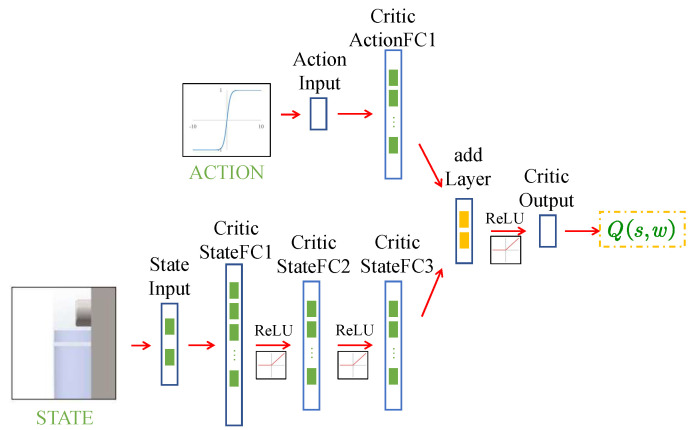
The diagram of Q network.

**Figure 5 micromachines-13-00458-f005:**
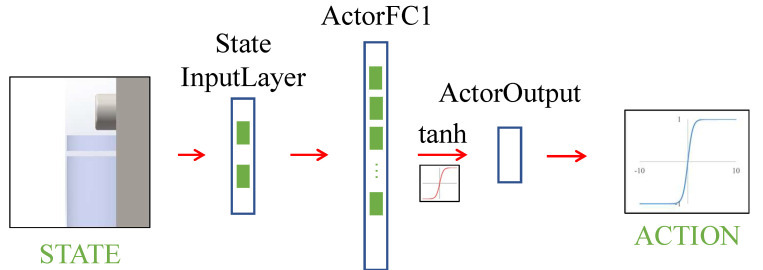
The diagram of policy network.

**Figure 6 micromachines-13-00458-f006:**
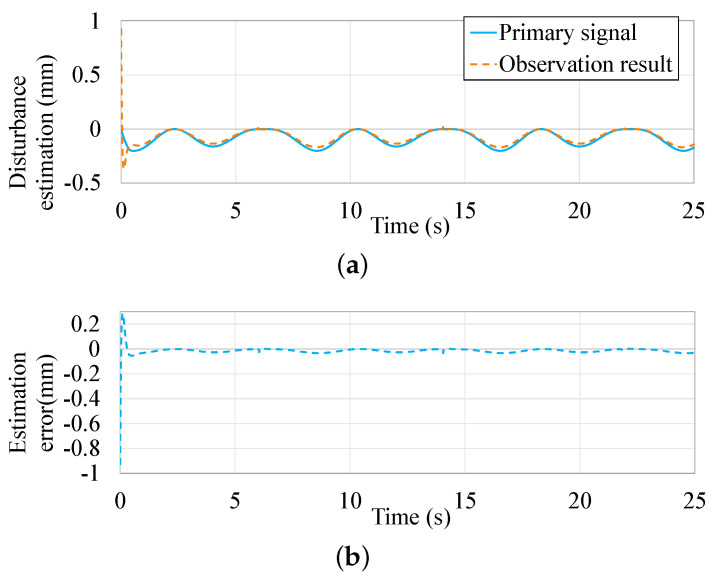
Observation result of ASMDO with $d_{2}$. (**a**) Observing result based on the ASMDO. (**b**) Observing error based on the ASMDO.

**Figure 7 micromachines-13-00458-f007:**
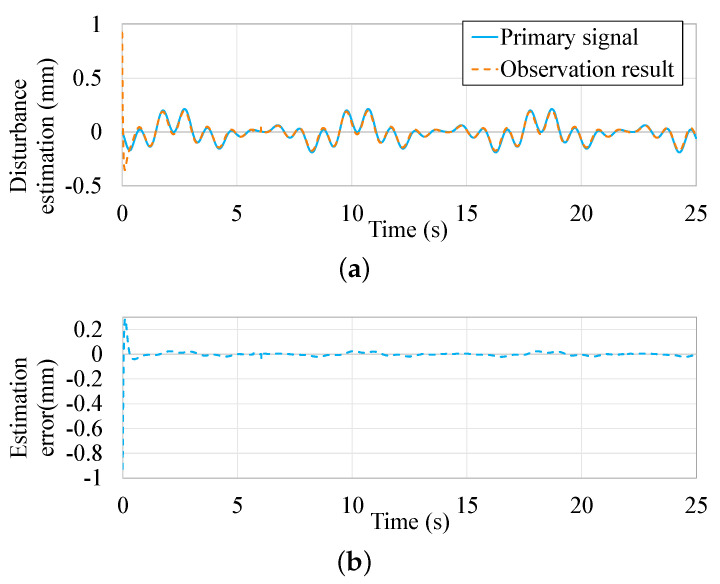
Observation result of ASMDO with $d_{1}$. (**a**) Observing result based on the ASMDO. (**b**) Observing error based on the ASMDO.

**Figure 8 micromachines-13-00458-f008:**
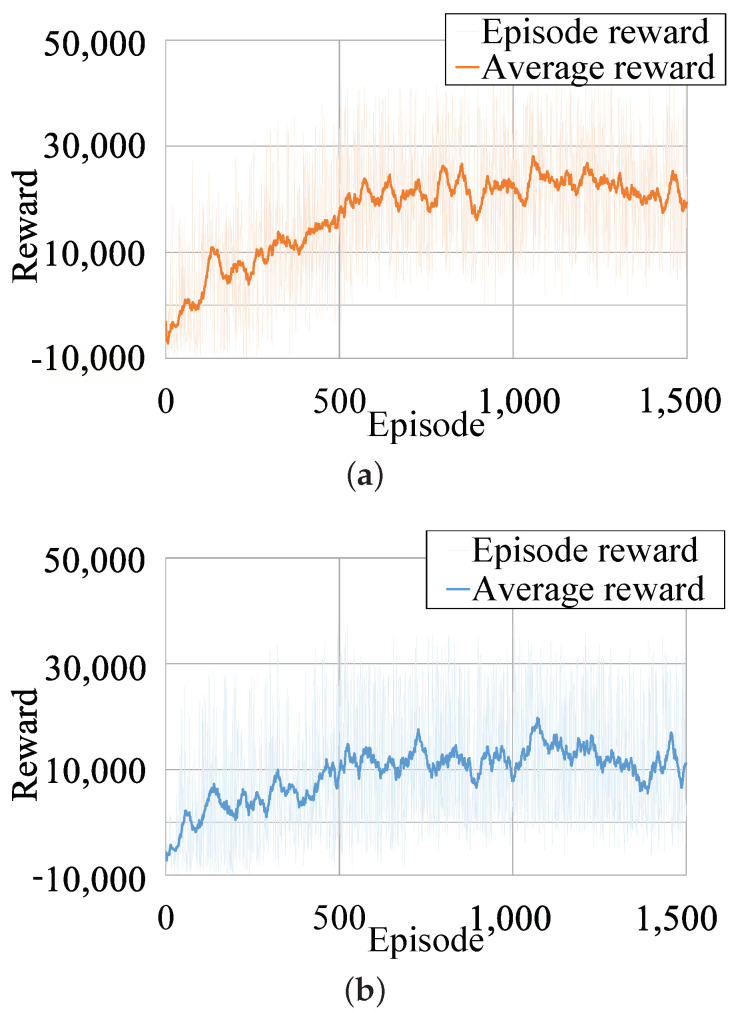
The training rewards of two RL schemes. (**a**) The training rewards generated by DDPG-ID. (**b**) The training rewards generated by DDPG.

**Figure 9 micromachines-13-00458-f009:**
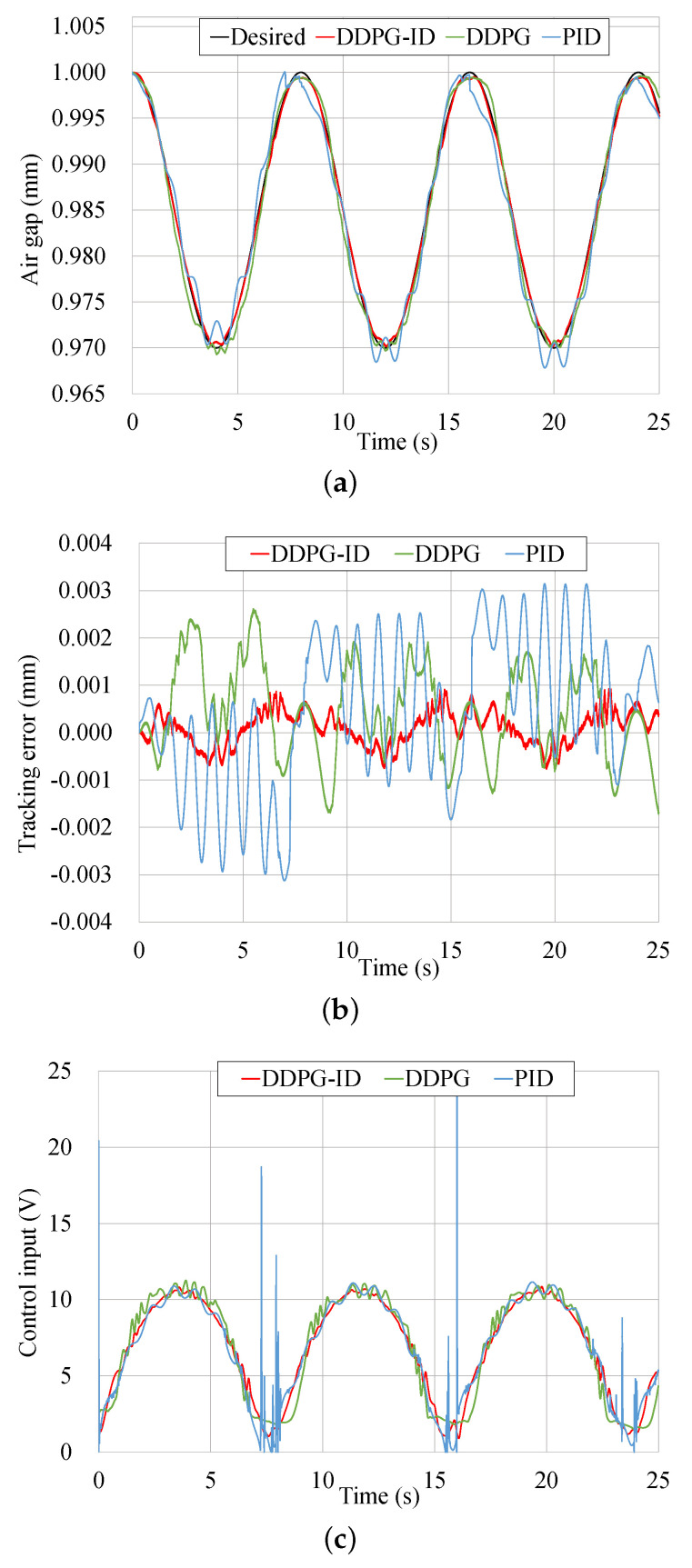
Tracking results comparison of the waved trajectory. (**a**) Tracking results comparison based on three control schemes. (**b**) Tracking error comparison based on three control schemes. (**c**) Control input comparison based on three control schemes.

**Figure 10 micromachines-13-00458-f010:**
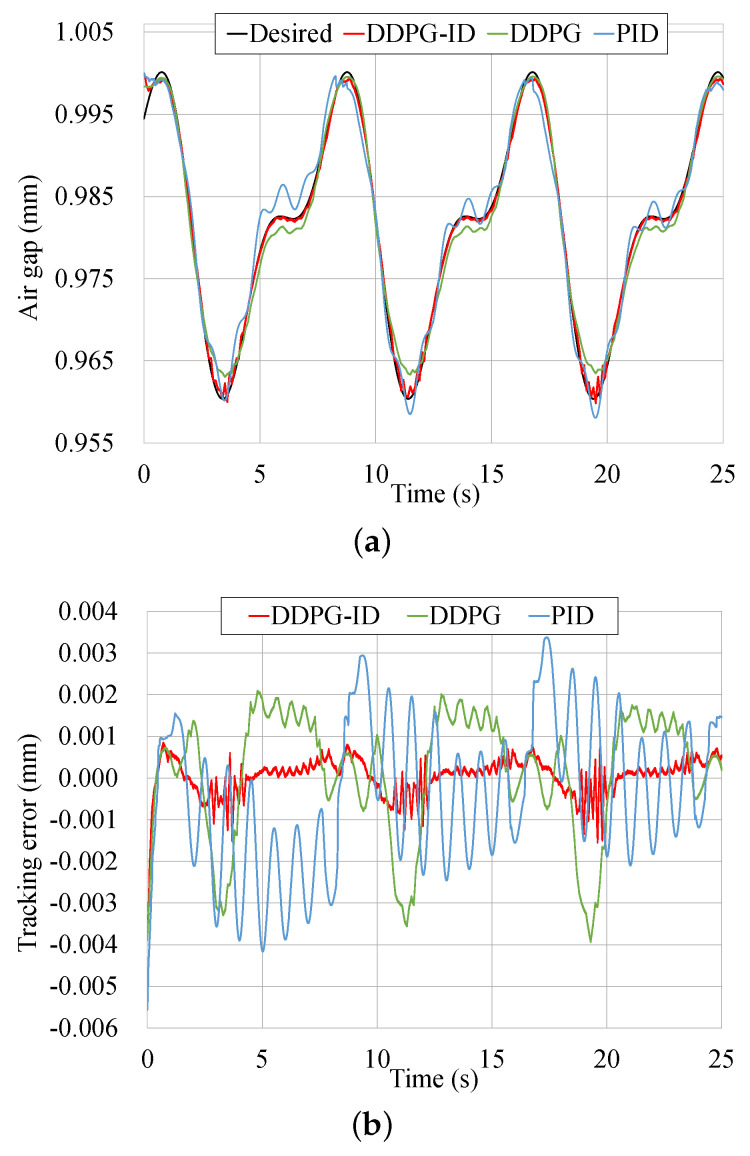
Tracking results comparison of the periodic trajectory. (**a**) Tracking results comparison based on three control schemes. (**b**) Tracking error comparison based on three control schemes. (**c**) Control input comparison based on three control schemes.

**Figure 11 micromachines-13-00458-f011:**
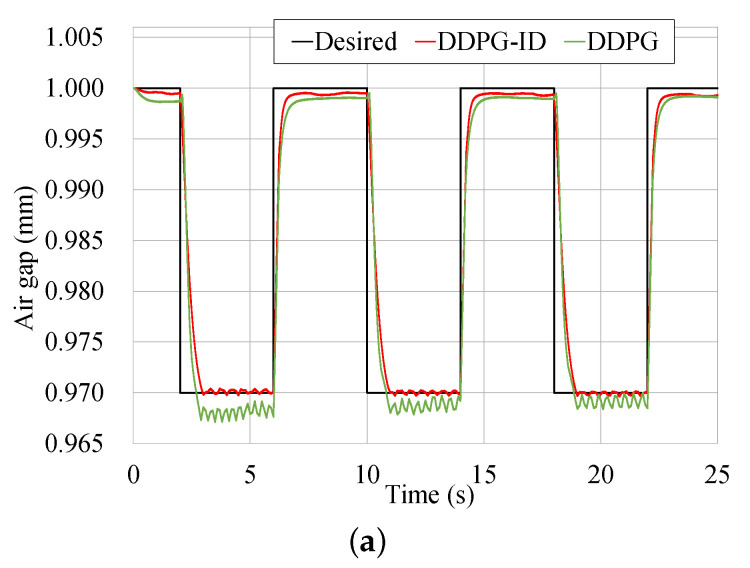
Tracking results comparison of the step trajectory. (**a**) Tracking results comparison based on two control schemes. (**b**) Tracking error comparison based on two control schemes. (**c**) Control input comparison based on two control schemes.

**Figure 12 micromachines-13-00458-f012:**
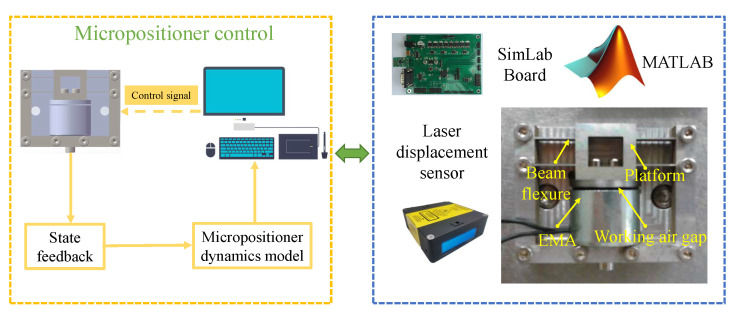
The schematic diagram of experiment system.

**Figure 13 micromachines-13-00458-f013:**
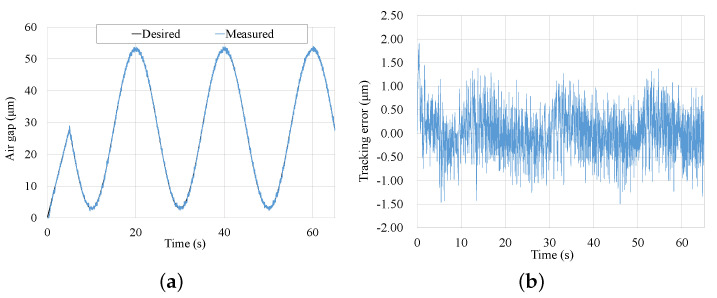
Tracking results of the waved trajectory. (**a**) Tracking result of desired trajectory. (**b**) Tracking error of desired trajectory.

**Figure 14 micromachines-13-00458-f014:**
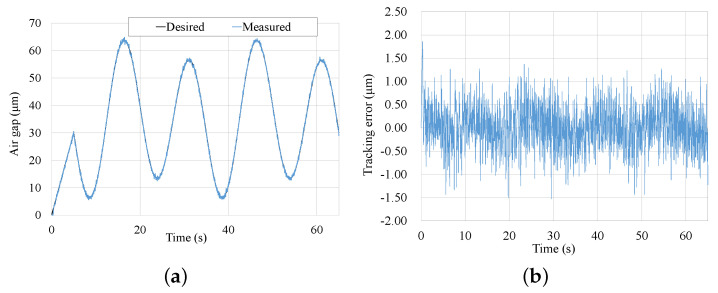
Tracking results comparison of the step trajectory. (**a**) Tracking result of desired trajectory. (**b**) Tracking error of desired trajectory.

**Table 1 micromachines-13-00458-t001:** Comparison of different control algorithms.

Method	Advantages	Disadvantages
PID control	Simple design structure Easy to implementation	Mainly used in linear systems Requirement of full-state feedback Lack of adaptivity
SMC control	Simple design structure Easy to implementation High robustness	Excessive chattering effect Lack of adaptivity
Adaptive control	Lower initial cost Lower cost of redundancy High reliability and performance	Stability is not treated rigorously High gain observes needed Slow convergence
Backstepping control	Global stability Simple design structure Easy to be integrated	Low anti-interference ability Sensitive to system models Lack of adaptivity
RL control	No need of accurate model Improved control performance High adaptivity	Poor anti-interference ability Easy to generate state error

**Table 2 micromachines-13-00458-t002:** Q network parameters.

Network Layer Name	Number of Nodes
StateLayer	5
CriticStateFC1	120
CriticStateFC2	60
CriticStateFC3	60
ActionInput	1
CriticActionFC1	60
addLayer	2
CriticOutput	1

**Table 3 micromachines-13-00458-t003:** Policy network parameters.

Network Layer Name	Number of Nodes
StateLayer	5
ActorFC1	30
ActorOutput	1

**Table 4 micromachines-13-00458-t004:** Parameters of the micropositioner model.

Notation	Value	Unit
$L_{1}$	13.21	H
$L_{0}$	0.67	H
*a*	$1.11\times 10^{-5}$	m
*R*	43.66	Ω
*c*	$8.83\times 10^{-5}$	${\mathrm{Nm}}^{2}\;A^{-2}$
*k*	$1.803\times 10^{N5}$	${\mathrm{Nm}}^{-1}$
*m*	0.0272	Kg

**Table 5 micromachines-13-00458-t005:** Training parameters of DDPG-ID and DDPG.

Hyperparameters	Value
Learning rate for actor $\varphi_{1}$	0.001
Learning rate for critic $\varphi_{2}$	0.001
Discount factor γ	0.99
Initial exploration ε	1
Experience replay buffer size ψ	100,000
Minibatch size *M*	64
Max episode ϖ	1500
Soft update factor τ	0.05
Max exploration steps *T*	250 (25 s)
Time step $T_{s}$	0.01 s
Intergal gain α	0.01
Differential gain β	0.001

**Table 6 micromachines-13-00458-t006:** Tracking errors comparison of different controllers in the waved trajectory.

	RMSE	MAX	MEAN
DDPG-ID	$3.658\times 10^{-4}$	$4.758\times 10^{-4}$	$1.003\times 10^{-4}$
DDPG	$1.093\times 10^{-3}$	$2.615\times 10^{-3}$	$4.414\times 10^{-4}$
PID	$1.654\times 10^{-3}$	$3.144\times 10^{-4}$	$3.104\times 10^{-4}$

**Table 7 micromachines-13-00458-t007:** Tracking errors comparison of different controllers in the periodic trajectory.

	RMSE	MAX	MEAN
DDPG-ID	$4.272\times 10^{-4}$	$8.471\times 10^{-4}$	$5.404\times 10^{-5}$
DDPG	$1.545\times 10^{-3}$	$3.102\times 10^{-3}$	$1.610\times 10^{-4}$
PID	$1.923\times 10^{-3}$	$3.376\times 10^{-3}$	$3.311\times 10^{-4}$

**Table 8 micromachines-13-00458-t008:** Tracking errors comparison of different controllers in the step trajectory.

	RMSE	MAX	MEAN
DDPG-ID	$4.612\times 10^{-3}$	0.02953	$6.938\times 10^{-4}$
DDPG	$5.279\times 10^{-3}$	0.02986	$1.437\times 10^{-3}$

## Data Availability

Not applicable.

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
