# Peer review of "Adaptive Sliding Mode Disturbance Observer and Deep Reinforcement Learning Based Motion Control for Micropositioners"

_micromachines, 2022, doi:10.3390/mi13030458_

Round 1

Reviewer 1 Report

Dear Authors,

This paper combines two highly appealing and extremely effective Adaptive Sliding Mode Disturbance Observer and Deep Reinforcement Learning based Robust Motion Control controllers. The quality of the figures is very good and also the paper is well organized. Although the experimental results of this paper were difficult, the authors performed admirably. However, the experimental and theoretical findings have fundamental challenges that need to be addressed.

Overall, this paper's subject is intriguing, and it has some publishable information. In addition, the presentation of the results in terms of the study objectives was effective. The following enhancements, however, could be considered.

1- After combining two methods a key aspect is the stability study of the overall closed-loop system in presence of uncertainties is not addressed carefully.

2- The work done has a strong foundation for proving the method. However, I must point out that it is not possible to refer to a robust control in an article without thoroughly evaluating the system output in the presence of various system uncertainties, theoretically and experimentally.

3- There are a few typos in the text such as ReLU instead of relu that need to be corrected.

4- It is critical to note that the actions taken regarding the dimensions and sizes of the micro-positioners discussed in this article have nothing to do with reality. A typical micro-position, for example, has a mass of around 0.001 grams, not the 27 grams mentioned in this article. Furthermore, when the system dimensions are tiny, all control charts, including system tracking, will reach their ultimate value in around 0.1 seconds, and it is preferable for such systems to analyze them in the same range.

5- The authors should refer to the most recent and relevant published works (2021 & 2022) to highlight the contributions of the present work. Dear authors, for example, are urged to utilize new robust control, Adaptive Sliding Mode and disturbance rejection theory in micropositioner systems references, such as the following references on control techniques in the field of micropositioners systems.

Ruan, W., Dong, Q., Zhang, X., & Li, Z. (2021). Friction Compensation Control of Electromechanical Actuator Based on Neural Network Adaptive Sliding Mode. Sensors, 21(4), 1508.

Nguyen, M. H., Dao, H. V., & Ahn, K. K. (2022). Adaptive Robust Position Control of Electro-Hydraulic Servo Systems with Large Uncertainties and Disturbances. Applied Sciences, 12(2), 794.

Gharib, M. R., Koochi, A., & Ghorbani, M. (2021). Path tracking control of electromechanical micro-positioner by considering control effort of the system. Proceedings of the Institution of Mechanical Engineers, Part I: Journal of Systems and Control Engineering, 235(6), 984-991.

Salehi Kolahi, M. R., Gharib, M. R., & Heydari, A. (2021). Design of a non-singular fast terminal sliding mode control for second-order nonlinear systems with compound disturbance. Proceedings of the Institution of Mechanical Engineers, Part C: Journal of Mechanical Engineering Science, 235(24), 7343-7352.

Author Response

Thanks to Reviewer 1 for the detailed comments,we have revised the manuscript carefully. The details are in the attached PDF file.

Reviewer 2 Report

The authors present the article entitled “Adaptive Sliding Mode Disturbance Observer and Deep Reinforcement Learning based Robust Motion Control for Micropositioners”. However, in its current form, it is not possible to extend my recommendation for publication by the following concerns:

The manuscript is found with a lot of grammatical and typographical errors. The authors are suggested to go through the manuscript thoroughly and proofread for grammatical and typographical errors. Please use technical language.        

Line 26: Please add et al when citing Fei

Include quantitative values in the abstract to highlight the findings.

Line 84: change “letter” for “manuscript”

Lines 67-69: Rewrite this sentence; it is hard to read.

Section 3: I recommend changing the section name according to its content.

Captions of Figures 2 and 3: Please give a brief description of Figures a) and b).

Lines 126-128: It seems that Figures 2 and 3 are part of the results section. Also, the analysis of Figures Fig. 2 and 3 in these lines is poor.

Section 4 and 5 could be merged in a single section called: Results

Include a table that compares the findings of the work vs the already reported in the state of the art.

In line 23-24 it is threatened the motion control approaches reported in the state of the art, it can be interesting to discuss the following ones: a new methodology for a retrofitted self-tuned controller with open-source fpga; a pid-type fuzzy logic controller-based approach for motion control applications; fpga-based architecture for sensing power consumption on parabolic and trapezoidal motion profiles; a new seven-segment profile algorithm for an open-source architecture in a hybrid electronic platform.

In the last section, Include quantitative results and mention future works.

Please add a nomenclature in order to present the acronyms and variables used in the manuscript.

Author Response

In view of the comments made by the reviewer, we have revised the manuscript carefully. The details are given in the attached PDF file.

Round 2

Reviewer 1 Report

First of all, thank you to the respected authors of the paper for providing comprehensive responses to the questions. Next, despite some minor flaws in the paper, it has been accepted for publishing in the journal without concern.

Yours Sincerely